# The Relationship and Influencing Factors between Endangered Plant *Tetraena mongolica* and Soil Microorganisms in West Ordos Desert Ecosystem, Northern China

**DOI:** 10.3390/plants12051048

**Published:** 2023-02-25

**Authors:** Zhangkai Liu, Congwen Wang, Xuejun Yang, Guofang Liu, Qingguo Cui, Tuvshintogtokh Indree, Xuehua Ye, Zhenying Huang

**Affiliations:** 1State Key Laboratory of Vegetation and Environmental Change, Institute of Botany, Chinese Academy of Sciences, Beijing 100093, China; 2University of Chinese Academy of Sciences, Beijing 100049, China; 3Botanic Garden and Research Institute, Mongolian Academy of Sciences, Ulaanbaatar 13330, Mongolia

**Keywords:** *Tetraena mongolica* community, desert ecosystem, soil properties, soil microorganisms, biodiversity, interaction

## Abstract

Soil microorganisms play crucial roles in improving nutrient cycling, maintaining soil fertility in desert ecosystems such as the West Ordos desert ecosystem in Northern China, which is home to a variety of endangered plants. However, the relationship between the plants–microorganisms–soil in the West Ordos desert ecosystem is still unclear. *Tetraena mongolica,* an endangered and dominant plant species in West Ordos, was selected as the research object in the present study. Results showed that (1) there were ten plant species in the *Tetraena mongolica* community, belonging to seven families and nine genera, respectively. The soil was strongly alkaline (pH = 9.22 ± 0.12) and the soil nutrients were relatively poor; (2) fungal diversity was more closely related to shrub diversity than bacterial and archaeal diversity; (3) among the fungal functional groups, endomycorrhizal led to a significant negative correlation between shrub diversity and fungal diversity, because endomycorrhizal had a significant positive effect on the dominance of *T. mongolica*, but had no significant effect on other shrubs; (4) plant diversity had a significant positive correlation with the soil inorganic carbon (SIC), total carbon (TC), available phosphorus (AVP) and available potassium (AVK). This study revealed the effects of soil properties and soil microorganisms on the community structure and the growth of *T. mongolica* and provided a theoretical basis for the conservation of *T. mongolica* and the maintenance of biodiversity in desert ecosystems.

## 1. Introduction

China has the largest desert ecosystem in the world, with more than 1.92 million square kilometers of desert ecosystem, accounting for about one-fifth of the land area, and is one of the countries with the most serious desertification disasters [1]. The particularity of desert plant communities makes them form extremely sensitive and fragile ecosystems adapted to the environment, which will be difficult to recover once damaged. West Ordos of Inner Mongolia is a unique region of desertified steppe/steppe desert in Northern China. Even though the ecological environment is harsh, this region is an important distribution area of plant endemic genera [2]. Some rare and endangered plants, such as *Tetraena mongolica* (Zygophyllaceae), *Ammopiptanthus mongolicus* (Leguminosae), and *Helianthemum songaricum* (Cistaceae), are distributed in the West Ordos desert ecosystem. However, due to the change in land use type, industrial pollution, and other human activities, as well as the stress of the harsh environment, the distribution area of these rare plants is gradually shrinking [3].

As an endangered but dominant shrub species in the desert of West Ordos desert ecosystem, *T. mongolica* plays an important role in maintaining and improving the fragile ecological environment of the desert and is also conducive to the balance of regional ecology and the maintenance of biodiversity [4,5]. Nowadays, researchers have carried out a large number of conservation studies on *T. mongolica*, mainly focusing on its physiological processes [6], genes [7,8], and potential survival suitable zone [9]. However, these studies may overlook the role of soil microorganisms in its protection.

Soil microorganisms are an important component of the ecosystem and play a crucial role in regulating the ecosystem functions and attributes, such as improving nutrient cycling, maintaining soil fertility, and carbon sequestration [10,11,12]. Soil microorganisms also play an important role in the coexistence of species in plant communities and the maintenance of biodiversity [13,14,15,16]. Soil microorganisms can regulate plant diversity by improving nutrient availability via degrading plant litter or residue, and enhancing nutrient uptake efficiency (e.g., mycorrhizal fungi) of plants via mutualisms [11,17]. The higher the biodiversity of soil microorganisms, the more beneficial substances they can provide and the more beneficial effects on plants and soil from the ecological mechanism [18]. Moreover, the functional diversity of soil microorganisms is beneficial to ecosystem stability. For example, fungi may promote ecosystem stability by increasing the drought resistance and resilience of plants under drought stress [19], and the antagonism between mycorrhizal fungi and pathogenic fungi can also moderate the negative effects of pathogenic fungi on ecosystem stability [20,21,22].

In recent years, researchers attempted to explore the links between plant and soil microbial diversity in different natural ecosystems [23,24,25], however, the results of these studies were inconsistent, especially in plant communities of natural habitats [26]. Some results showed that plant diversity was positively correlated with fungal diversity [24,27], while others showed that plant diversity was not strongly correlated with the diversity of fungi, bacteria, and archaea [23]. In addition, these findings suggest that the association between plant diversity and soil fungal diversity is stronger than that between plant diversity and soil bacterial and archaeal diversity [27]. Similarly, mycorrhizal fungi, which facilitate nutrient uptake by forming symbiotic relationships with plants, have positive or negative feedback on plant diversity in different regions. For example, mycorrhizal fungi have been shown to increase plant diversity by promoting seedling establishment and enhancing the competitiveness of non-dominant plants [28,29,30], but some other studies showed that mycorrhizal fungi can reduce plant diversity under certain circumstances, particularly in ecosystems where dominant plants are highly mycorrhizal dependent and derive the greatest benefit from mycorrhizal fungi [31,32]. Therefore, more empirical evidence is needed to elucidate the relationship between plant–soil feedback and soil microbial communities.

Understanding the complex plant–soil–soil microbial interactions at the local scale will help us to achieve more comprehensive conservation of endangered plants in the West Ordos desert ecosystem and better understand the mechanisms that maintain plant diversity at the local scale. Nowadays, few studies explored the relationship between *T. mongolica* and soil microorganisms through greenhouse control experiments. For example, the structure of bacterial and fungal communities in the rhizosphere of *T. mongolica* [33], the destruction of bacterial community and soil properties in the rhizosphere of *T. mongolica* by coal mining [34], and the *T. mongolica* seedlings inoculated with AM fungi could adapt to drought stress by improving antioxidant enzyme activities [35]. However, how soil microorganisms and soil physicochemical properties affect the plant diversity of the *T. mongolica* community in the natural environment is still unclear.

In this study, we investigated the community characteristics, soil microbial community structure and function, and habitat soil properties of the endangered plant *T. mongolica* in the West Ordos desert ecosystem, through field investigation and high-throughput sequencing, to explore the relationships among plants–soil–soil microorganisms. The objectives of this study were (1) to explore how soil microorganisms and soil properties affect plant diversity of the *T. mongolica* community and the growth of *T. mongolica*; (2) to illuminate the relationship among plant–soil–soil microorganisms in the desert ecosystem, and (3) to provide a theoretical basis for the protection of *T. mongolica* and the maintenance of biodiversity.

## 2. Results 

### 2.1. Characteristics of Plant, Soil, and Soil Microbial Communities in T. mongolica Community

There were ten different plant species found in the *T. mongolica* community, including four shrubs and six herbs, belonging to nine genera and seven families. *T. mongolica* was taken as the dominant species of shrubs with an important value of 0.56, and *Tripolium vulgare* was taken as the dominant species of herbs with an important value of 0.65 (Table 1). The richness of plants was 5.3 ± 1.2, the average coverage was 24 ± 7%, the aboveground biomass of shrubs was 0.60 ± 0.20 kg/m^2^, that of herbs was 8.8 ± 6.8 g/m^2^, the weight of litter was 17.7 ± 10.7 g/m^2^, and standing litter was 7.2 ± 3.2 g/m^2^. The diversity index of shrubs was higher than that of herbs (Table 2).

The soil was strongly alkaline (pH = 9.22 ± 0.12) in the *T. mongolica* community, total potassium concentration (TK) was 19.35 ± 0.72 g/kg, total phosphorus concentration (TP) was 350.34 ± 22.57 mg/kg, soil inorganic carbon concentration (SIC) was 7.65 ± 3.12 g/kg, available phosphorus concentration (AVP) was 3.94 ± 1.20 mg/kg, available potassium concentration (AVK) was 136.79 ± 23.13 mg/kg, total nitrogen concentration (TN) was 0.34 ± 0.048 g/kg, and total carbon concentration (TC) was 10.62 ± 3.83 g/kg.

The richness index of fungi was 759.2 ± 81.19, that of bacteria was 2450.8 ± 135.59, and that of archaea was 176.8 ± 27.82, the diversity index of bacteria was higher than that of fungi and archaea (Table 2). Among the functional groups of fungi, saprotroph was the highest (39.46%), followed by pathogen and ectomycorrhizal (Appendix A). 

Among the phylum composition of soil microorganisms, Ascomycota was the most common fungi, followed by Basidiomycota and Glomomycota; the most common bacteria were Actinobacteria, followed by Proteobacteria and Chloroflexi; Thaumarchaeota was the most common archaea, followed by Euryarchaeota. In the genus-level composition of soil microorganisms, the main genus of fungi was Chaetomium, followed by Aspergillus and Acrophialophora. Most genera of bacteria have not been classified yet (71.79%), while Rubrobacter, Geodermatophilus, and Microvirga were the main genera in the classified bacteria. Most genera of archaea had not been classified (77.68%) either, and Candidatus_Nitrososphaera, Haladaptatus, and Halarchaeum were the main genera in the classified archaea (Appendix A). 

### 2.2. Relationship and Influencing Factors of Aboveground and Belowground Biodiversity

Spearman correlation test showed that plant richness was positively correlated with SIC, TC, AVP, and AVK, and the Shannon diversity index of shrubs was positively correlated with TN. There was no significant relationship between plant diversity and soil microbial diversity. However, the shrub diversity index had a significant negative correlation with the fungal diversity index (Figure 1). Compared with SIC, AVK, and TC, the increase in AVP had the strongest effect (*R*^2^ = 0.897, *p* < 0.001) on plant diversity (Figure 2). 

The shrub diversity index had a significant negative correlation with the fungal diversity index, mainly due to the Mortierellomycota and Calcarisporiellomycota of fungal phyla. The plant diversity index was negatively correlated with Actinobacteria and positively correlated with Proteobacteria in bacteria. The plant diversity index showed a significant negative correlation with Euryarchaeota and a significant positive correlation with Thaumarchaeota in archaea. There were different positive and negative relationships between plant diversity and bacteria or archaea, which led to the unclear relationship between plant diversity and bacteria or archaea diversity (Figure 3).

There were significant negative correlations between the diversity indexes of fungi and SIC, AVP, TN, and TC, while there was a significant positive correlation between the richness indexes of fungi and bacteria and pH (Figure 1). Among them, TN had the greatest effect on fungal diversity (Figure 4). There was a significant negative correlation between Ascomycota and pH in fungi. Gemmatimonadetes of bacteria had a significant positive correlation with AVP, and Firmicutes had a significant positive correlation with TK (Figure 3).

There was a strong competitive relationship among soil microbial communities, and Mortierellomycota of fungi had a significant negative correlation with Proteobacteria of bacteria. There was also a significant negative correlation between Ascomycota and Basidiomycota in fungi, Actinobacteria and Proteobacteria in bacteria, and Euryarchaeota and Thaumarchaeota in archaea (Figure 3).

### 2.3. Relationship and Influencing Factors between the Community of T. mongolica and Fungal Functional Groups 

There was no significant relationship between plant diversity and the richness index of fungal functional groups, but there was a significant negative correlation between shrub diversity index and endomycorrhizal, and endomycorrhizal had a significant negative correlation with TC, SIC, and TN content (Figure 5), and SIC content had the greatest effect on the richness of endomycorrhizal (Figure 4). There was a significant positive correlation between the importance value of *T. mongolica* and endomycorrhizal fungi, other functional groups of fungi had insignificant correlations with community diversity or characteristic values of *T. mongolica*, and other shrubs had insignificant correlation with endomycorrhizal fungi (Figure 6).

## 3. Discussion

The complex plants–soil–soil microbial relationship is one of the hot issues in ecological research [36,37]. It is important to understand the relationships among the three for ecosystem management and protection, especially for desert ecosystems with simple community structures and poor soil nutrients [19]. The present study provided new evidence about the relationships among plants–soil–soil microorganisms in the West Ordos desert ecosystem, in Northern China. It showed that soil nutrients, especially AVP, had a significant positive relationship with plant diversity, a negative relationship with fungi diversity, and no significant relationship with bacteria and archaea diversity; fungal diversity was more closely related to shrub diversity than bacteria and archaea diversity; and among the fungal functional groups, endomycorrhizal had a significant positive effect on *T. mongolica*, but had no significant effect on other shrubs.

Compared to other studies of desert ecosystems or other ecosystems [38,39,40,41,42,43,44], the *T. mongolica* community in this study had fewer plant species, lower plant productivity, lower soil nutrient, and the soil pH was strongly alkaline. The different physicochemical properties of the soil will have different effects on vegetation, thus affecting plant diversity, and the growth process of vegetation is the continuous adaptation and improvement of plants to the soil environment [45] Some studies had shown that there was a significant positive correlation between plant diversity and soil nutrients [46], while others had shown that increased soil nutrients reduce plant diversity, which was caused by the competitive relationship between plants [47]. Our study showed that there was a significant positive correlation between the plant diversity of the *T. mongolica* community and the soil content of SIC, TN, TC, and especially AVP. The increase in plant diversity was conducive to the production of more litter into the soil layer, and the continuous accumulation of soil nutrients could also promote the increase in plant diversity. Therefore, the application of fertilizer (especially P fertilizer) during the conservation of *T. mongolica* would help to maintain plant diversity in the community.

Soil pH can indirectly affect plant growth by affecting soil physicochemical properties (soil structure, and soil nutrient availability), or directly affect plant growth by affecting substance absorption, root development, and seed germination and plant growth requires a suitable soil pH environment [18,48]. In the present research, the soil was strongly alkaline with a pH value of more than 9. Our study found no significant relationship between plant diversity and pH in the *T. mongolica* community, different from another study that found that there was a significant negative correlation between pH and plant diversity in desert ecosystems [49]. Yet there was a significant positive relationship between plant richness and AVP, and since alkaline soil has a direct influence on P availability, it seems that the high pH value in the present study might be an important factor for the relict plant *T. mongolica*. Further studies were needed to expound on how soil pH affects plant growth, distribution, and even seed germination of *T. mongolica*.

Soil microorganisms play a more critical role in desert ecosystems with poor soil fertility, since rich and stable soil microbial diversity is conducive to maintaining soil fertility, preventing and controlling diseases, and promoting plant growth [11]. Soil microorganisms, especially bacteria and fungi, can regulate plant diversity by degrading plant litter or debris to improve nutrient availability, and by increasing nutrient uptake efficiency by plants through mutualism [17]. Changes in plant diversity can change the distribution of plant litter and root exudates to underground ecosystems, which in turn directly or indirectly affect soil ecological processes and lead to changes in soil microbial communities [15,19]. No significant correlation between plant diversity and bacterial diversity had also been found in studies of the northern grasslands of China and temperate grasslands worldwide [23,50]. This is consistent with our results that the relationship between plant diversity and bacterial diversity was not significant. However, there was a significant negative correlation between the plant diversity index and the Actinobacteria of bacteria, and a significant positive correlation with Proteobacteria of bacteria in the present study. It seems that different bacterial species had different responses to plant diversity, which weakened the relationship between plant diversity and bacterial diversity.

Archaea have been shown to be important in various terrestrial ecosystems [51], but no studies have been conducted to date to reveal how archaea are affected by plant diversity [27]. Our study showed that there was a significant negative correlation between plant diversity and Euryarchaeota of archaea, and a significant positive correlation with Thaumarchaeota, which might have weakened the relationship between plant diversity and archaea diversity. It was reported that the settlement of *T. mongolica* could significantly change the composition and function of the archaea community and improve the archaea diversity in the rhizosphere soil of *T. mongolica* [52]. Fungi had stronger effects on the plant community and the growth of *T. mongolica* than bacteria or archaea in the present study.

There was a significant negative correlation between the shrub diversity index and the fungal diversity index, and it was further discovered that the significant negative correlation between the shrub diversity index and the fungal diversity index was due to Mortierellomycota and Calcarisporiellomycota. Through the division of fungal functional groups, our results showed that there was a significant negative correlation between the endomycorrhizal and shrub diversity, and there was also a negative correlation between the arbuscular mycorrhizal fungi (AMF) and shrub diversity (r = −0.64, *p* > 0.05). Mycorrhizal fungi form symbiotic relationships with plant roots to enhance resource complementarity by providing nutrients unavailable to plant roots [13], and mycorrhizal fungi (especially AMF) had shown a positive correlation with plant diversity in most studies [30,50], but some studies had found a negative correlation between mycorrhizal fungi and plant diversity, such as dominant plants have a high degree of dependence on mycorrhizal fungi and through the mycorrhizal fungi to obtain the maximum benefit of the ecosystem [31,32], which resulted in dominant plants occupying more resources of other plants and decreasing plant diversity. Our results showed a significant positive correlation between the endomycorrhizal and the important values of *T. mongolica*, indicating that *T. mongolica* might have a selection specificity for the endomycorrhizal, this means that the increase in endomycorrhizal was conducive to the improvement of the dominance of *T. mongolica*. However, more evidence is needed to reveal how *T. mongolica* highly depends on endomycorrhizal fungi and takes up more resources. In addition, the increase in *T. mongolica* dominance may lead to a decrease in the shrub diversity index, which might explain the negative correlation between endomycorrhizal and the shrub diversity index.

Our study showed that soil C and N contents were closely related to *T. mongolica* and endomycorrhizal fungi. This reflected the complex relationship among shrub, soil, and soil microorganisms in the *T. mongolica* community. Mycorrhizal fungi are resistant to disease and drought and provide a series of limiting nutrients to plants, including nitrogen, phosphorus, copper, iron, and zinc, in exchange for carbon [13], which can provide more resources for plant growth and help maintain higher plant diversity, and improve the distribution of plant litter and root exudates to underground ecosystems [19]. There was a study that reported that with the increase in soil microbial richness and plant diversity, the C and N content in soil was increased in the alpine ecosystem [26]. However, our study found a significant negative correlation between the abundance of endomycorrhizal fungi and soil TC, SIC, and TN contents, which might be due to different soil sample sources (rhizosphere or non-rhizosphere soil), the soil in this study was taken from the habitat soil, while the endomycorrhizal fungi existed in the plant rhizosphere. When the soil nutrients increase, the competitive relationship between microorganisms might be intensified, and the endomycorrhizal fungi might also accumulate in the rhizosphere of the plant. It was still unclear which of the plants, soil, or soil microorganisms played a dominant role and how the endomycorrhizal utilize C and N, which needs further research. Notably, endomycorrhizal fungi had been shown to enhance P uptake for plants, especially in soils with poor nutrient availability and strong alkalinity, P could be mineralized and thus not directly absorbed by plants [53,54]. As mentioned above, this study had showed that the increase in available P was beneficial to the improvement of the plant diversity. Therefore, if the dynamic balance among the growth of *T. mongolica*, endomycorrhizal fungi, and soil AVP, TC, and TN contents was well maintained, it would be beneficial to the growth of *T. mongolica* and the maintenance of plant diversity. The results of this study provided empirical evidence for a better understanding of plant–soil–soil microbial interactions in desert ecosystems, and this would facilitate a better understanding of the relationship between *T. mongolica* and its habitat soil environment, which would provide theoretical support for the conservational application of endangered plant species, such as the development of microbial fertilizer and ex situ conservation.

## 4. Materials and Methods 

### 4.1. Study Area 

The study area was located in the Western Ordos National Nature Reserve, Inner Mongolia, China (106°53′1.34″ E, 40°4′54.43″ N, 1080 a.s.l.), with an altitude of about 1080 m. The West Ordos National Nature Reserve has the continental monsoon climate characteristics of warm temperate zone with large temperature difference between day and night (average daily temperature difference of 12.8–13.3 °C), less dry and less rain, and long sunshine hours. The annual sunshine duration was 3047–3227 h, the annual average air temperature was 7.8–8.1 °C, the annual extreme high temperature was 39.4 °C, the annual extreme ground temperature was −32.6 °C, the annual precipitation was 162–272 mm (concentrated in June–August), the annual potential evaporation was 2470–3481 mm, which was about 9–20 times of the precipitation, and the annual average relative humidity was 43% [55]. According to the Chinese vegetation classification system, the vegetation type in this area is temperate shrub desert, and the main plant community types include *T. mongolica* community, *Ammopiptanthus mongolicus* community, *Potaninia mongolica* community, *Convolvulus tragacanthoides* community, and *Helianthemum soongoricum* community [56].

### 4.2. Plant Community Survey

In August 2019, a permanent plot (100 m × 100 m) was set up in the study area for *T. mongolica* community (Figure 7A). According to the mechanical distribution method, thirteen shrub quadrats (10 m × 10 m) were set up in the plot. Two herb quadrats (1 m × 1 m) were set up in each shrub quadrat, and two destructive sampling quadrats (1 m × 1 m) were set up near the outside of the shrub quadrats (Figure 7B). The species, number, height, and crown width (coverage) of all the plants in the shrub quadrat and the herb quadrat as well as the total coverage of the quadrat were recorded in the survey. All herbs on the ground were cut off in the destructively sampled quadrats, which were then taken back to the laboratory for drying and weighing and recorded as herb biomass. Litter and stumpage in the destructively sampled quadrats were separately selected and taken back to the laboratory for drying and weighing, which were recorded as litter weight (LW) and stumpage weight (SLW). A stainless steel cylindrical sampler (with a diameter of 3 cm) was used for repeated sampling three times at random places (0–20 cm deep) in the destructive sampling quadrats. After the soil and other impurities were screened out with a 2 mm mesh, the fine roots were selected and taken back to the laboratory for drying and weighing, which were recorded as the underground biomass of herbs.

### 4.3. Investigation and Measurement of Soil Properties

In the destructive sampling quadrats, stainless steel cylindrical sampler (with a diameter of 3 cm) was used three times for repeated sampling at random places (0–20 cm deep) in the quadrats. Plant residues and gravels were removed through a 2 mm sieve and mixed into a mixed sample. A total of 13 mixed soil samples were collected from the permanent plots of *T. mongolica* for analysis. These soil samples were taken back to the laboratory for air-drying and then used to measure the physical and chemical properties of the soil. Three areas (within 1 m of the plant base or the vicinity of the plant) were randomly selected in each shrub sample plot. 

The pH of air-dried soil samples was measured using a pH meter (PB-10, Sartorius, Germany). Soil inorganic carbon (SIC) was determined by solid-state infrared carbon–sulfur analyzer (multi EA4001, Analytik-Jena AG, Jena, Germany). The total carbon (TC) and total nitrogen (TN) in the soil were determined with the Elemental Analyzer (Vario MACRO cube CHNOS Elemental Analyzer, Elementar Analysensysteme GmbH, Hanau, Germany). Soil total phosphorus (TP), available potassium (AVP), and total potassium (TK) were determined by inductively coupled plasma atomic emission spectrometry (iCAP 6300 ICP-OES Spectrometer, Thermo Fisher, USA). Available phosphorus (AVK) in soil was determined by colorimetry using a UV–visible spectrophotometer (UV-2550, UV–visible SPECTRO Photometer, Shimadzu, Japan).

### 4.4. Soil Microbial Sample Collection, DNA Sequencing, and Bioinformatics Analysis

The stainless steel cylindrical sampler (with a diameter of 3 cm) was used to collect the soil samples of soil microorganisms in the surface soil (0–20 cm). Three soil samples were randomly collected from each shrub quadrat. Animal and plant residues and gravel were removed through a 2 mm mesh and mixed into a mixed sample. Finally, 13 soil samples were collected from soil microorganisms, which were then refrigerated, transported, and stored (−20 °C) for subsequent sequencing in the laboratory.

Soil DNA was extracted from a 0.5 g homogeneous soil sample using the PowerSoil DNA Isolation Kit (MoBio Laboratories, Carlsbad, CA) as per the manual. Genomic DNA was examined for purity and quality on a 1% agarose gel. To evaluate the abundance of soil bacteria, archaea, and fungi, we amplified the V3–V4 hypervariable region of the 16S ribosomal RNA (16S) gene of bacteria and archaea using forward primers 338F (5′-ACTCCTGAGGGGGCAGGAG-3′) and 806R (5′-GGACTACVGGGTWTCTAAT-3′) [57]. Fungal internal transcribed spacer (ITS) using forward primer ITS1F (5′-CTTGGTCATAGAAGAAGTAA-3′) and reverse primer ITS2 (5′-TGCGTTCTTCAGATGC-3′) [58]. For each soil sample, a 10-position bar code sequence was added at the 5′ end of the forward and reverse primers. A polymerase chain reaction (PCR) was performed on a Mastercycler gradient (Eppendorf, Hamburg, Germany) in a 25 μL reaction volume consisting of 12.5 μL of 2 × TAQ PCR Master Mix, 3 μL bovine serum albumin (2 ng/μL), 2 μL primers (5 M), 2 μL template DNA, and 5.5 μL double distilled water (ddH_2_O). Cycle parameters were 94 °C for 5 min, followed by 28 and 32 cycles for 16 s, 94 °C for 30 s, 55 °C for 30 s, and 72 °C for 60 s, respectively, followed by a 7 min extension at 72 °C. Three PCR products per sample were combined to reduce PCR bias at reaction levels. The PCR products were purified using a QIAquick gel extraction kit (QIAGEN, German) according to the manufacturer’s instructions and sequenced on the Illumina MiSeq 300 PE platform (Illumina, San Diego, CA, USA) at Allwegene Technology, Beijing, using real-time PCR quantitation. The original sequence was read followed by trimming with mothur [59]. After quality control, more than 95% of the original sequence readings were retained. If the sequence was less than 200 bp and the mass fraction was low, the sequence (≤20) was discarded, including ambiguous bases or an incomplete match with the primer sequence and the barcode label. QIIME was used to analyze the dataset. The Operational Classification Unit (I) is the operational definition used to classify closely related individuals in phylogenetic or population genetic studies. The sequences were aggregated into OTUs at the level of ninety-seven percent similarity using UPARSE. Each I was classified using SILVIA and UNITE [60]. Microbial diversity was generally calculated based on the number of OTUs obtained by clustering in the high-throughput sequencing. In this study, the species diversity of soil microorganisms was calculated based on the number of OTUs per soil sample. To avoid errors in sequencing results, OTUs ≥ 10 were used for analysis during the analysis of this study.

### 4.5. Statistical Analysis

All data analyses in this section are based on the R language (version 4.11). The functional groups of fungal communities were predicted by FUNGuild. The Shannon diversity index of plant and soil microorganisms was calculated using the vegan package. All environmental data, such as soil property data for TC, TN, and TP, were log10 log-transformed. The correlation (spearman coefficient) and significance among plants, soil, and soil microorganisms were analyzed using the corrplot package, and the correlation heat map was drawn. The importance of a plant in the community is expressed by the important value of species, which is calculated by the following formula: important value of species = (Relative Frequency + Relative Density + Relative Coverage)/3.

## 5. Conclusions

This study provided clear evidence that soil nutrients, especially AVP, had a significant positive relationship with plant diversity in *T. mongolica* communities, contributing to the maintenance of higher plant diversity. Fungi diversity was more closely related to shrub diversity than bacteria and archaea diversity, and among the fungal functional groups, endomycorrhizal fungi had a significant positive effect on *T. mongolica* but had no significant effect on other shrubs. Plants with different nutrient acquisition strategies had different interactions (such as symbiosis) with soil microorganisms (especially endomycorrhizal fungi). These findings could help us to better understand the mechanisms of maintaining plant diversity at local scales and provide a more comprehensive theoretical basis for the conservation process of endangered plants.

## Figures and Tables

**Figure 1 plants-12-01048-f001:**
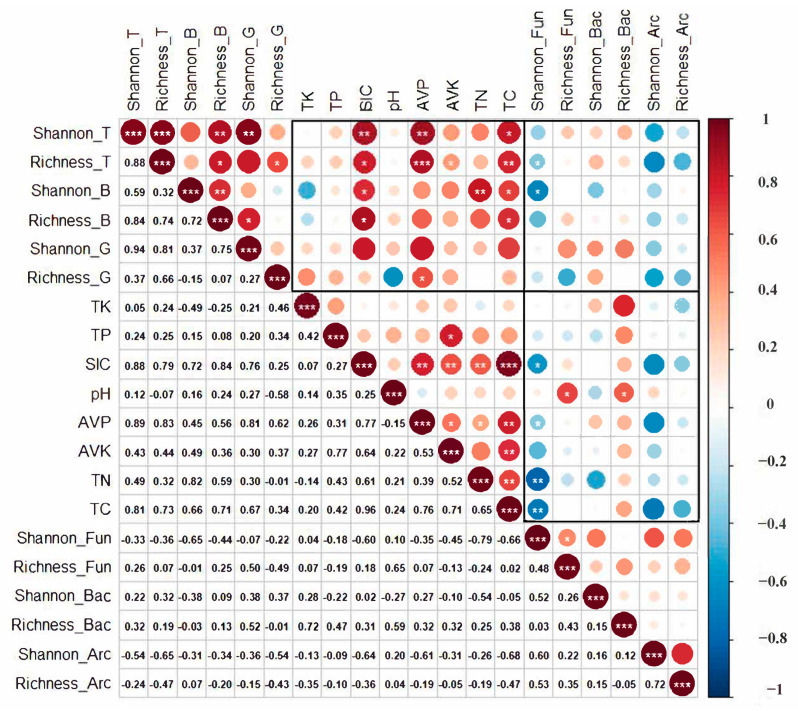
The correlation heat map of aboveground and belowground diversity in *Tetraena mongolica* community. Different colors represent different positive and negative correlations and the magnitude of correlations. The specific values of correlations are shown in the lower part of the figure. “*” means significant difference, that is, *p* < 0.05; “**” means the difference is very significant, that is, *p* < 0.01; “***” means the difference is extremely significant, that is, *p* < 0.001. In the figure, the suffix “_T” represents the plant as a whole, the suffix “_B” represents the bush, the suffix “_G” represents the herb, the suffix “_Fun” represents the fungus, the suffix “_Bac” represents the bacterium, and the suffix “_Arc” represents the archaea.

**Figure 2 plants-12-01048-f002:**
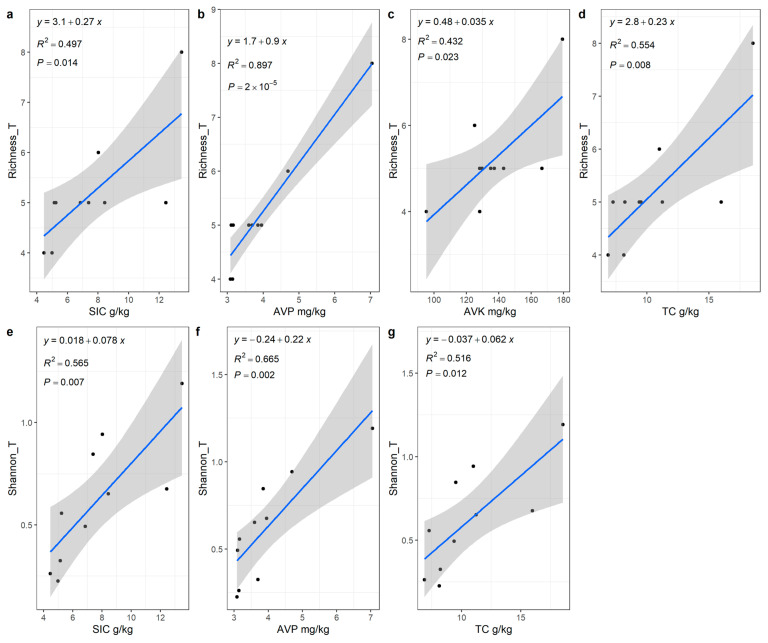
Linear regression analysis of plant diversity and soil content in *Tetraena mongolica* community. (**a**–**d**) shows the linear regression of whole plant richness index with SIC (**a**), AVP (**b**), AVK (**c**) and TC (**d**), respectively. (**e**–**g**) shows the linear regression of Shannon diversity index and SIC (**e**), AVP (**f**) and TC (**g**), respectively. In the figure, the suffix “_T” represents the whole plant.

**Figure 3 plants-12-01048-f003:**
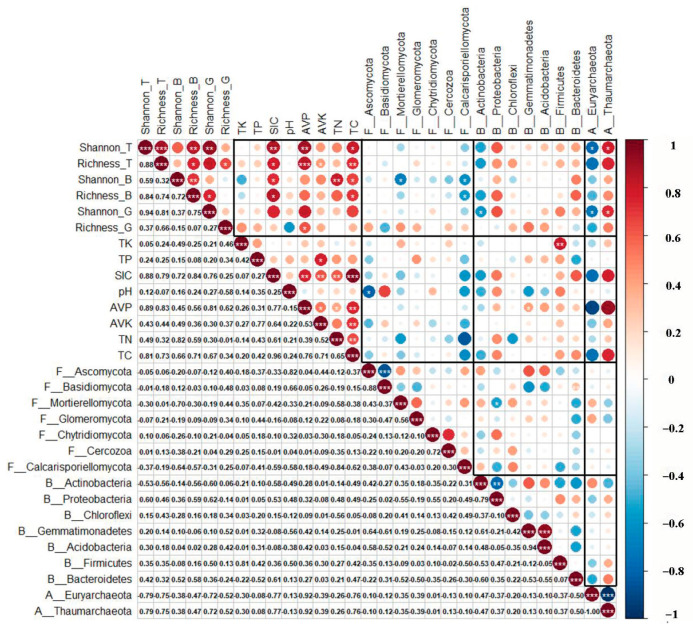
The correlation between plant diversity and relative abundance of soil microbial phylum in *Tetraena mongolica* community. Different colors represent different positive and negative correlations and the magnitude of correlations. The specific values of correlations are shown in the lower part of the figure. “*” means significant difference, that is, *p* < 0.05; “**” means the difference is very significant, that is, *p* < 0.01; “***” means the difference is extremely significant, that is, *p* < 0.001. In this figure, the suffix “_T” represents the total plant as a whole, the suffix “_B” represents the bush, the suffix “_G” represents the herb, the prefix “F_” represents the fungus, the prefix “B_” represents the bacterium, and the prefix “A_” represents the archaea.

**Figure 4 plants-12-01048-f004:**
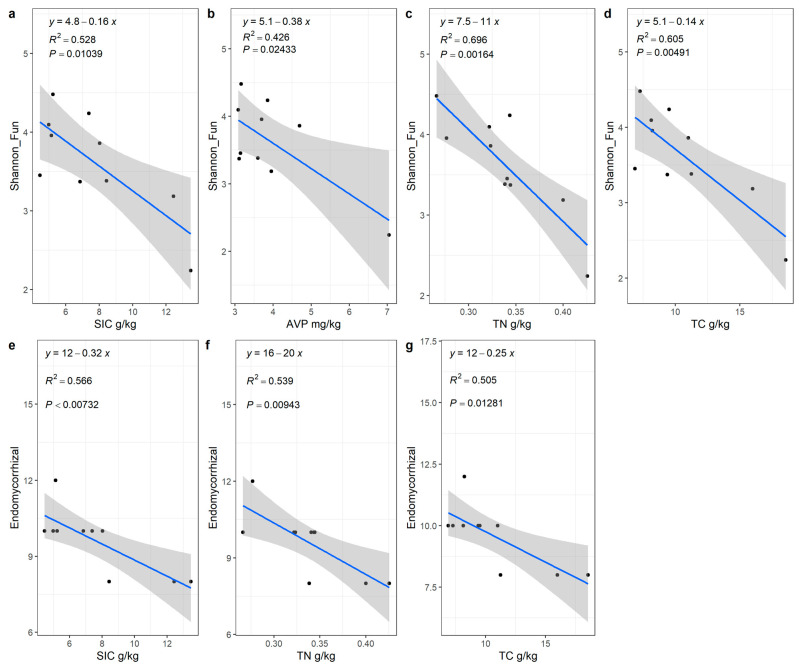
Linear regression analysis of fungal diversity, endomycorhizal richness, and soil content in *Tetraena mongolica* community. (**a**–**d**) shows the linear regression of Fungal Shannon diversity index and SIC (**a**), AVP (**b**), TN (**c**) and TC (**d**), respectively. (**e**–**g**) shows the linear regression of the richness index of endomycorrhizal fungi with SIC (**e**), TN (**f**) and TC (**g**), respectively.

**Figure 5 plants-12-01048-f005:**
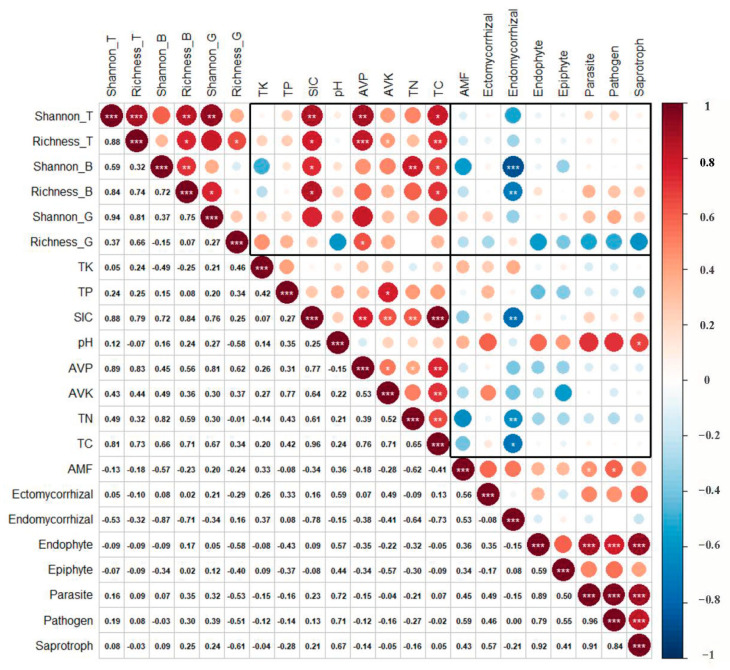
The correlation between aboveground diversity and fungal functional richness in *Tetraena mongolica* community. Different colors represent different positive and negative correlations and the magnitude of correlations. The specific values of correlations are shown in the lower part of the figure. “*” means significant difference, that is, *p* < 0.05; “**” means the difference is very significant, that is, *p* < 0.01; “***” means the difference is extremely significant, that is, *p* < 0.001. In the figure, the suffix “_T” represents the whole plant, the suffix “_B” represents the bush, and the suffix “_G” represents the herb.

**Figure 6 plants-12-01048-f006:**
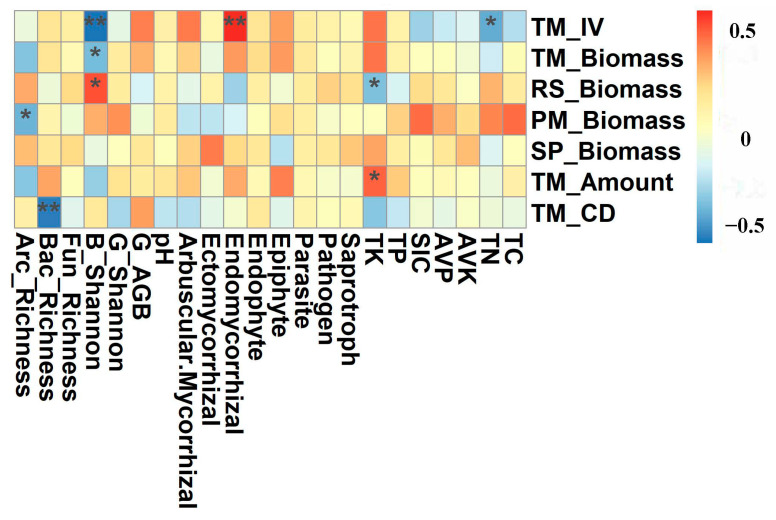
The correlation between the characteristics of *Tetraena mongolica* and soil characteristics and soil microbial community. Different colors represent different positive and negative correlations and the magnitude of correlations. The specific values of correlations are shown in the lower part of the figure. “*” means significant difference, that is, *p* < 0.05; “**” means the difference is very significant, that is, *p* < 0.01. In the figure, TM_IV is the important value of *T. mongolica*, TM_Biomass is the biomass of *T. mongolica*, RS_Biomass is the biomass of *Reaumuria songarica*; PM_Biomass is the biomass of *Potaninia mongolica*; SP_Biomass is the biomass of *Salsola passerina*; TM_Amount is the number of plants including *T. mongolica*, TM_CD is the richness of Archaea Including Arc_Richness, Bacteria including Bac_Richness and Fungi including Fun_Richness. B_Shannon is shrub diversity index, G_Shannon is herb diversity index, and G_AGB is herb aboveground biomass.

**Figure 7 plants-12-01048-f007:**
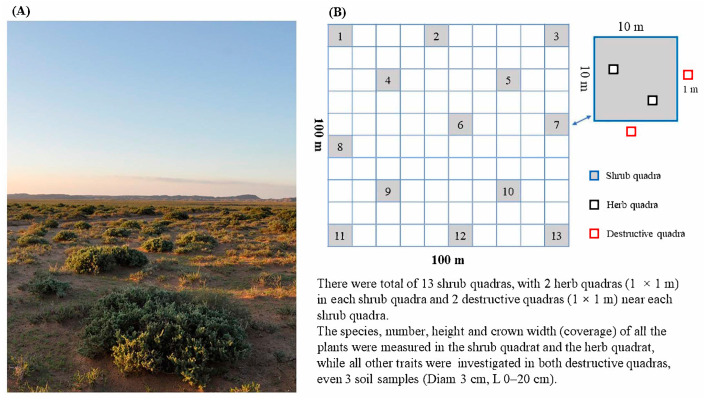
(**A**) The habitat of *T. mongolica* in the study site. (**B**) The quadrat setup for this study.

**Table 1 plants-12-01048-t001:** Plant list and species importance value in *Tetraena mongolica* community.

Plant Species	Plant Life Form	Family	Genus	Average Height (cm)	Coverage by Species	Density (/hm^2^)	Importance Value
*Tetraena mongolica*	Shrub	Zygophyllaceae	*Tetraena*	33.95	21.46	55.6	0.56
*Reaumuria songarica*	Shrub	Tamaricaceae	*Reaumuria*	38.08	7.82	18.6	0.29
*Potaninia mongolica*	Shrub	Rosaceae	*Potaninia*	20.27	1.64	15.5	0.13
*Salsola passerina*	Shrub	Amaranthaceae	*Salsola*	21.54	0.27	2.6	0.05
*Tripolium vulgare*	Herb	Asteraceae	*Tripolium*	19.8	0.755	4705	0.65
*Cleistogenes chinensis*	Herb	Poaceae	*Cleistogenes*	3.0	0.225	425	0.18
*Echinops sphaerocephalus*	Herb	Asteraceae	*Echinops*	16.4	0.345	180	0.098
*Cleistogenes squarrosa*	Herb	Poaceae	*Cleistogenes*	2.2	0.075	115	0.027
*Peganum harmala*	Herb	Zygophyllaceae	*Peganum*	9.75	0.25	10	0.025
*Limonium sinense*	Herb	Plumbaginaceae	*Limonium*	9	0.05	5	0.015

**Table 2 plants-12-01048-t002:** Index of plant diversity and soil microbial diversity of *Tetraena mongolica* community.

	Type	Richness Index	Shannon Diversity Index
Plant	Shrub	2.7 ± 0.82	0.62 ± 0.35
	Herb	2.5 ± 0.85	0.47 ± 0.21
	Shrub + Herb	5.2 ± 1.14	0.56 ± 0.27
Soil Microbial	Fungi	759.2 ± 81.19	3.63 ± 0.65
	Bacteria	2450.8 ± 135.59	5.35 ± 0.08
	Arachaea	176.8 ± 27.82	2.45 ± 0.23

## Data Availability

The data presented in this study are available in this article and Appendix A.

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
