# Peer review of "The Relationship and Influencing Factors between Endangered Plant Tetraena mongolica and Soil Microorganisms in West Ordos Desert Ecosystem, Northern China"

_plants, 2023, doi:10.3390/plants12051048_

Round 1
Reviewer 1 Report
Soil microorganisms play crucial roles in improving nutrient cycling, maintaining soil fertility in desert ecosystem. In this manuscript, Tetraena mongolica was selected to explore the relationship between plant and soil microbial in West Ordos. It is an interesting paper and the subject certainly falls within the general scope of Plnats journal. However, I have some concerns about the work presented in the MS.
1. Title need to revise to make it more impact. This MS mainly studied on the relationship of plants- microorganisms -soil, however, "microorganisms" and "soil" did not included in title.
2. The Abstract is not a good writing and needs to be substantially improved, and it suggest to add some exact results with data or some description.
3. The introduction: it is not well organized. More recent literature related to the investigated topic has been published so they should be included in your introduction. Also, it should be enhanced with more related papers from top-ranked journals. The last paragraph should be rewritten by highlighting the main contribution of current work compared with existing literature. In addition, the purpose of this study was not only the relationship between plants and soil microorganisms, but also the relationship and influencing factors between the community of plants and soil chemical- and physical- properties, therefore, it suggest to introduce the references about soil chemical- and physical- properties and plant's diversity.
4. The conclusion section should be considerably improved. In addition, the authors are advised to clearly summarize the main results, as well as explain the significance of their results using quantitative reasoning. The originality and innovations of the study should be clarified and highlighted. Here, the last sentence "Our results illustrated the relationship between plant diversity and different soil microbial taxa in the T. mongolica community, and emphasized soil characteristics and soil microorganisms should be paid attention to during the conservation of T. mongolica." is obvious, and it is of nonsense.
5. The English of the paper is OK, but can be improved for gaining higher attention worldwide.
In summary, the MS has its merits, it could be considered for publication with Major revision.
Author Response
Thank you very much for your helpful comments. Please check the attachment for point-by-point response.

Reviewer 2 Report
Dear authors,
Thanks for the interesting article. I proposed adding a photograph of the site (for people like me who don't know this plant-desert association) and an illustration of the sampling. Then I highlighted an inconsistency in the results, with endomycorrhizal fungi sometimes being negatively correlated with variables with which they should instead be positively correlated. It is a question of correcting the conclusions also in the light of a recent publication of which I have reported the authors and details (annexed PDF). Thanks again and best wishes for your career.

Author Response

(The authors gave the same response as above.)

Round 2
Reviewer 1 Report
It appears that authors have worked hard to improve the whole text, and the manuscript is acceptable in its current form.
Author Response
Dear Reviewer,
Thank you very much for your valuable suggestions which have greatly improved this article.
With best wishes.